

# Developing Spring Wheat in the Noah-MP LSM (v4.4) for Growing Season Dynamics and Responses to Temperature Stress

Zhe Zhang[1,2], Yanping Li[1,2], Fei Chen[3], Phillip Harder[1,4], Warren Helgason[1,4], James Famiglietti[1,2],
Prasanth Valayamkunnath[3], Cenlin He[3], Zhenhua Li[1,2]

[1]Global Institute for Water Security, University of Saskatchewan, 11 Innovation Blvd, Saskatoon, SK, S7N 3H5, Canada
[2]School of Environment and Sustainability, University of Saskatchewan, 117 Science Place, Saskatoon, SK, S7N 5C8, Canada
[3]Research Applications Laboratory, National Center for Atmospheric Research, P.O. Box 3000, Boulder, CO, USA
[4]College of Engineering, University of Saskatchewan, 57 Campus Dr, Saskatoon, SK, S7N 5A9, Canada

*Correspondence to*: Yanping Li (yanping.li@usask.ca)

**Abstract.** The US Northern Great Plains and the Canadian Prairies are known as the world's breadbaskets for its large spring wheat production and exports to the world. It is essential to accurately represent spring wheat growing dynamics and final yield and improve our ability to predict food production under climate change. This study attempts to incorporate spring wheat growth dynamics into the Noah-MP crop model, for a long time period (13-year) and fine spatial scale (4-km). The study focuses on three aspects: (1) developing and calibrating the spring wheat model at point-scale, (2) applying a dynamic planting/harvest date to facilitate large-scale simulations, and (3) applying a temperature stress function to assess crop responses to heat stress amid extreme heat. Model results are evaluated using field observations, satellite leaf area index (LAI), and census data from Statistics Canada and the US Department of Agriculture (USDA). Results suggest that incorporating a dynamic planting/harvest threshold can better constrain the growing season, especially the peak timing and magnitude of wheat LAI, as well as obtain realistic yield compared to prescribing a static province/state-level map. Results also demonstrate an evident control of heat stress upon wheat yield in three Canadian Prairies Provinces, which are reasonably captured in the new temperature stress function. This study has important implications for estimating crop productions, simulating the land-atmosphere interactions in croplands, and crop growth's responses to the raising temperatures amid climate change.

## Plain Language Summary

Crop models incorporated in earth system models are essential to accurately simulate crop growth processes on Earth's surface and agricultural production. In this study, we aim to model the spring wheat in the Northern Great Plains, focusing on three aspects: (1) develop the wheat model at point-scale; (2) apply dynamic planting/harvest schedules; (3) adopt a revised heat stress function. The results show substantial improvements and have great importance for agricultural production.





## 1 Introduction

Wheat is a widely grown temperate cereal and a major staple crop for global food security, ranked fourth among commodity crops, with a global production of 711 million tons. The Prairie Provinces in Canada (Alberta, Saskatchewan, Manitoba) and

the U.S. Northern Great Plains are known as the breadbasket of North America, producing spring wheat which is the first and third largest commodity crop in Canada and the U.S., respectively. At the same time, Canada and the U.S. account for approximately 20% of the global wheat export market, according to the US Department of Agriculture and Agriculture and Agri-Food Canada (ERS USDA report, Statistics Canada & AAFC).

Spring wheat is planted in late spring after snowmelt and soils have drained sufficiently to allow fieldwork and is harvested in

late summer to avoid early fall frost. The spatial and temporal variability of climate across the entirety of the spring wheat production area of the northern great plains means planting typically starts early April in southern portions and concludes by late May in the northern portions. Within the growing season, four major growing stages are identified for spring wheat, including seedling, emergence, anthesis, and grain-filling, largely aligned with the spring and summer weather. Thus, weather variations, especially temperature, play an important role in the environmental control at each growing stage. Accumulated

heat, calculated though growing degree day (GDD) accumulation, is an effective proxy to quantify crop stage.

Wheat is sensitive to high-temperature stress, whose negative impacts have been reported, particularly under higher end emission scenario (Lobell et al., 2011a; IPCC 2014; Qian et al. 2019; Agyeman et al., 2021). When the temperature is higher than the optimal temperature range, the impacts may be three-fold: (1) high temperature may close the leaf stomata, reducing the $CO_2$ absorption and transpiration, and increase photorespiration competition, reducing photosynthetic efficiency, (2)

damage the activity of enzymes in leaf chloroplast, (3) and fastens the phenological developments, especially the grain-filling stage for biomass accumulation. These processes will reduce crop photosynthesis at the physiological level and affect crop phenological development.

As such, it is important to understand and accurately represent growing dynamics and responses to heat stress for spring wheat, from plant physiology to its parameterization in earth system model applications. Previous studies have utilized statistical

regression models to connect agricultural production with weather inputs. Usually, the spatial region for each regression model is large and there is lack of detailed process understanding in these statistical studies (Qian et al. 2011; Carew et al., 2018). Ensemble process-based crop model studies (e.g., AgMIP study, Rosenzweig, et al., 2013; Jägermeyr et al., 2021) have applied statistical downscaled forcing from ensemble GCMs to demonstrate the impacts of climate change on crop production (Semenov and Shewry, 2018). Although studies provided important quantification the uncertainty range originating from GCM

forcing, RCP scenarios, and crop model parameterizations, this approach has simplified processes for complex energy, water, and carbon interactions occurring on the surface during crop growing seasons, including their potential feedback to the atmosphere.

There is an emerging trend for physical process-based crop models integrated within Earth System Models (ESMs) to specifically investigate dynamic crop growth and heat response under climate change (Levis 2014; McDermid et al., 2019). In



particular, The NoahMP crop model (Liu et al., 2016) integrates dynamic crop growth processes of two major crops, corn and soybean, into simulation of surface energy, water, and carbon fluxes (Niu et al., 2011; Yang et al., 2011), and can be further coupled with the Weather Research & Forecasting model (WRF, Skamarock et al., 2008), for regional climate simulations. In addition, Xu et al. (2019) incorporated an irrigation scheme based on soil moisture deficit and these two schemes were jointly tested in Zhang et al. (2020) for the U.S. Midwest Cornbelt and Mississippi River Valley with reasonable performance. As the

third largest crop planted in the US and Canada, the dynamic growing process of wheat is not yet developed in the NoahMP LSM, calling for better representation of wheat growing processes.

This study has three goals: (1) develop a dynamic wheat growth model in the Noah-MP crop from the Kenaston site in Saskatchewan; (2) conduct large regional wheat simulations for the Northern Great Plains and Canadian Prairies; and (3) address the temperature response function and investigate the impact of heat stress on crop yield in this region. The structure

of this paper is as follows: Section 2, Model and Data, will introduce the Noah-MP crop model and the necessary data used in this study. Section 3, Results, will present the results from three designed simulations: a single-point model, large regional simulations, and accounting for heat stress function. Section 4 provides a broad discussion of discrepancies between model results and evaluation datasets, planting/harvest practice in real agricultural management, and the temperature stress function. The final conclusions are discussed in Section 5.


## 2 Model and Data

### 2.1 Noah-MP crop model

In this study, we mainly used the inherent model structure in the Noah-MP crop model, added a new crop species, spring wheat, and developed new features, such as a dynamic planting/harvest date for regional application, and investigating the crop responses to high temperature. The Noah-MP LSM is widely used for modeling land surface processes, energy and water balance, and the land surface component coupled with regional weather and climate model (WRF). The crop model in Noah-MP was initially developed in Liu et al. (2016) to accommodate corn and soybean, two major crops grown in the U.S. Zhang et al. (2020) performed a joint crop and irrigation simulation for these two crop species in the U.S. for large regional-scale application.

The Noah-MP crop model has three main components, the photosynthesis-stomata component, the growing degree day (GDD) component, and the carbohydrate allocation component. The photosynthesis (PSN)-stomata component calculates the $CO_2$ assimilation and stomatal conductance, given environmental conditions, such as radiation, $CO_2$ level, temperature, moisture stress, and plant leaf area index (LAI). This photosynthesis-stomata component contains the key processes in which the crops are actively impacted by and respond to environmental conditions. The GDD component accumulates the daily GDD (Eq. 1) and determines the crop growing stages, according to temperature and GDD thresholds for each stage set in the parameter table. The carbohydrate allocation component partitions assimilated carbohydrates (converted from assimilated $CO_2$ from photosynthesis) to different parts (leaf, stem, root, and grain) of the plant structure, depending on the growing stage function, with feedback between LAI and photosynthesis. These three components together constitute complete plant growing processes in phenology, physiology, biogeophysics and biogeochemistry. Please see Appendix A for the full parameter table adjusted for spring wheat growth.

$$GDD = \begin{cases} 25; & (T > GDDCUT) \\ T - GDDBASE; \\ 0; & (T < GDDBASE) \end{cases}, \tag{1}$$

GDDTBASE (5 ℃) and GDDTCUT (30 ℃) are the commonly used base and cut-off temperature parameters for wheat GDD accumulation (Saiyed et al., 2009).

$$ACGDD = \sum GDD, \tag{2}$$

ACGDD is the accumulated daily GDD within the growing season. When ACGDD passes through growing stage thresholds, the crop would enter different stages, including emergence, initial vegetative stage, reproductive stage, maturation, and harvest. The threshold parameters determining these stages are presented in Appendix A.





## 2.2 Dynamic planting & harvest dates

The model spring wheat growing season is determined by the planting and harvest dates. For a starting point, they can be
prescribed from field records in single-point simulations. For large-scale simulations, two approaches are adopted in this study.
The first approach is to use the spatially varying values from the most common planting/harvest date for each state and province
in US and Canada, respectively (USDA NASS, 2010, province/state level static map). This static map is shown in Figure 1.

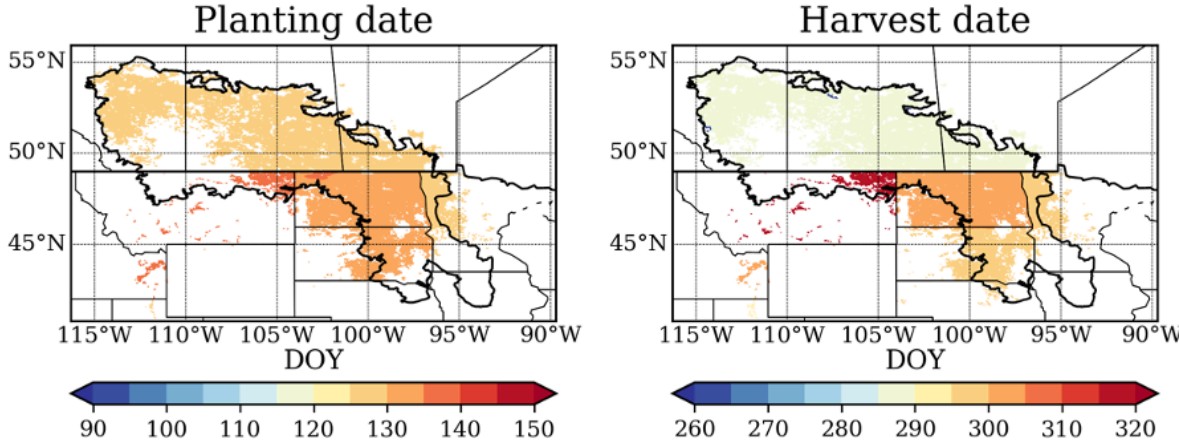

**Figure 1. Province/State-level planting & harvest map for DOY (day or year) in the Canadian Prairies and Northern Great Plains region.**

The second approach is introducing dynamic planting and harvest, based on meteorological input, most importantly,
temperature. Sacks et al. (2010) provided a global synthesis of planting and harvest dates across the globe and indicated that
the mostly likely planting temperature in the Canadian Prairies is around 10 ℃. Iizumi et al. (2019) attempted to model the
planting and harvest window for major global crops and demonstrated that environmental factors controlled by temperature
and precipitation, and by soil moisture and snowpack, play important roles in determining the planting and harvest timing.
Similar environmental thresholds are applied in this study for both planting and harvest:

$$TAVE_{(5)} > 10 \, , \tag{3}$$

Planting is triggered by 5-day running average temperature ($TAVE_{(5)}$) greater than 10 ℃.

$$ACGDD > 1500, \tag{4}$$

Harvest is triggered when accumulated GDD ($ACGDD$) passes 1500.



### 2.3 Temperature stress function

Regarding the plant physiological response to temperature, a stress function has been applied in the photosynthesis-stomata

subroutine in NoahMP crop, originally from Collatz et al. (1991):

$$f(TV) = (1.0 + exp\,((-2.2E05 + 710 * (TV + 273.16))/(8.314 * (TV + 273.16))))\,)^{-1} \quad , \tag{5}$$

$$V(TV) = f(TV) * Vcmx25 * (2.4)^{\frac{TV-25}{10}}, \tag{6}$$

Eq (5) represents the temperature stress function itself, vegetation canopy temperature ($TV$) is used in this equation, plotted as

the black dotted line in Figure 2(a). Eq (6) shows the combined temperature stress function with the temperature increase for

the *Vcmx25*, a parameter describing the rubisco capacity enzyme at temperature (25℃), which increases exponentially (base

at 2.4) with every 10 °C temperature increase. This exponential increase function is also known as the Q10 function (($2.4)^{\frac{T-25}{10}}$).

The combined effect of Q10 exponential increase and temperature stress $f(TV)$ is plotted in Figure 2(b). This combined effect

of temperature on rubisco capacity shows a one-peak function which is optimal at about 33 °C.

However, this temperature is much higher than the actual optimal temperature for wheat growth (25~30 °C), due to the

exponential increase from the Q10 function in Eq. 6. It is suggested that the rubisco parameter Vcmx display a decrease at

higher temperatures (Harley and Tenhunen 1991; Bernacchi et al., 2013) and that Eq. 5 and Eq.6 can be integrated together.

Wang and Engel proposed an alternative temperature stress function, from a synthesis of 29 widely used wheat models, also

known as the Wang-Engel equation (1998) (Eq. 7). This new equation is visualized as the blue line in Figure 2b and has a peak

near 27 °C, a more reasonable range compared to Eq. 6 (Collatz 1991). This heat stress function is adopted in the third

experiment in this study (Section 2.6).

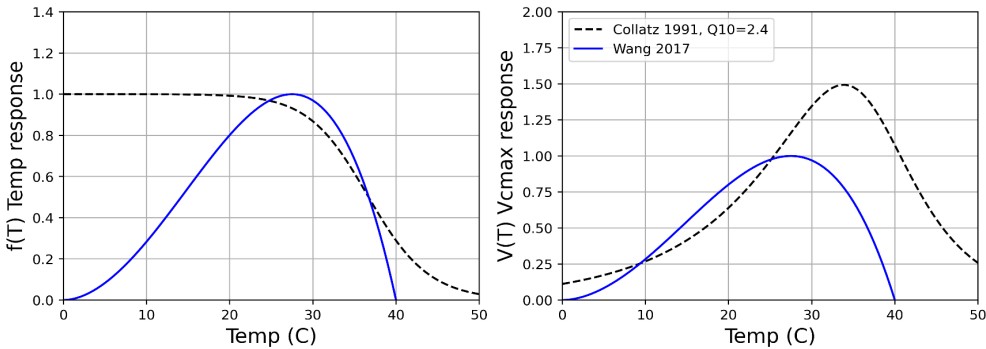

**Figure 2. Temperature response of the default Noah-MP PSN-stomata scheme (Collatz et al., 1991) and a new temperature response function (Wang et al., 2017) revised equation.**





### 2.4 Kenaston site

Observational data comes from sites in the Brightwater Creek watershed located 80-km south of Saskatoon, Saskatchewan, Canada. The Agricultural Water Futures project of the Global Water futures program (https://gwf.usask.ca/projects-facilities/all-projects/p3-ag-water-futures.php) since 2016 has collected agricultural water use and energy fluxes data for several crop species, including wheat, barley, canola, lentils, peas, and forages. These sites exhibit characteristics consistent with dryland agricultural production in the Canadian Prairies. Three site years of data for wheat specifically were collected in 2016 and 2019 at the SE13 site (51.389, -106.437) and 2019 at the SW30 site (51.420, -106.422). Soils range between silt-loam at SE13 and clay-loam at SW30 and limited topography and runoff tends to restrict surface water redistribution to local depressions. Observations are focused on quantifying land-atmosphere water and energy exchange, soil moisture dynamics, and crop growth metrics.

Data included in this study include turbulent fluxes, sensible (SH) and latent heat (LH), as observed with eddy covariance. Campbell Scientific CSAT3 sonic anemometers and EC150 gas analyzers or Irgason (an integrated CSAT and EC150) systems collected high frequency observations with data processing and QA/QC completed with default settings in Li-Cor EddyPro software. Gap filling and energy balance closure was completed with the REddyProc packages (Wutzler et al., 2018) in R to generate a continuous 30-minute time series of SH and LH observations. Soil moisture observations at 2 depths (5 and 20 cm) were collected with Stevens Hydraporobes. Due to salinity interactions, especially at the SE13 site, absolute values need to be treated with caution while relative dynamics are more meaningful to validate model dynamics. Crop growth metrics, above ground biomass and LAI, were collected during biweekly site visits. The biomass sampling protocol provides an average value for each date from the removal of all above ground biomass, subsequent oven drying, and weighing of between 6 and 12 samples with each sample having a 0.25 m$^2$ ground surface extent. LAI was sampled with a Decagon Accupar LP-80 ceptometer. The reported value comes from the average of 20 samples taken perpendicular and parallel to crop rows every 3 meters over a 30 meter transect adjacent to the eddy covariance station. Care was taken to ensure stable sky conditions over the course of observations within 2 hours of solar noon.



## 2.5 Regional data for agricultural management and model evaluation

We further expanded the single-point model to cover a large wheat-planting region across the U.S. and Canada, according to
the latest planting area data acquired from crop inventory data. The George Mason University CropScape dataset (USDA-
NASS CropScape) and the Annual Crop Inventory (AAFC ACI) from AAFC together provide a high-resolution (30-m) crop
frequency map for the past 10 years (Figure 2a). Statistics Canada and the United States Department of Agriculture (USDA)
also collect the planting/harvest areas at Census Agricultural Region (CAR) and county level, respectively (Figure 3b). Due to
their different units and the sizes of census regions, the results are presented at 1000 hectares and two color scales are used for
Canadian and the US to accommodate the wide ranges of values from these two data sources.

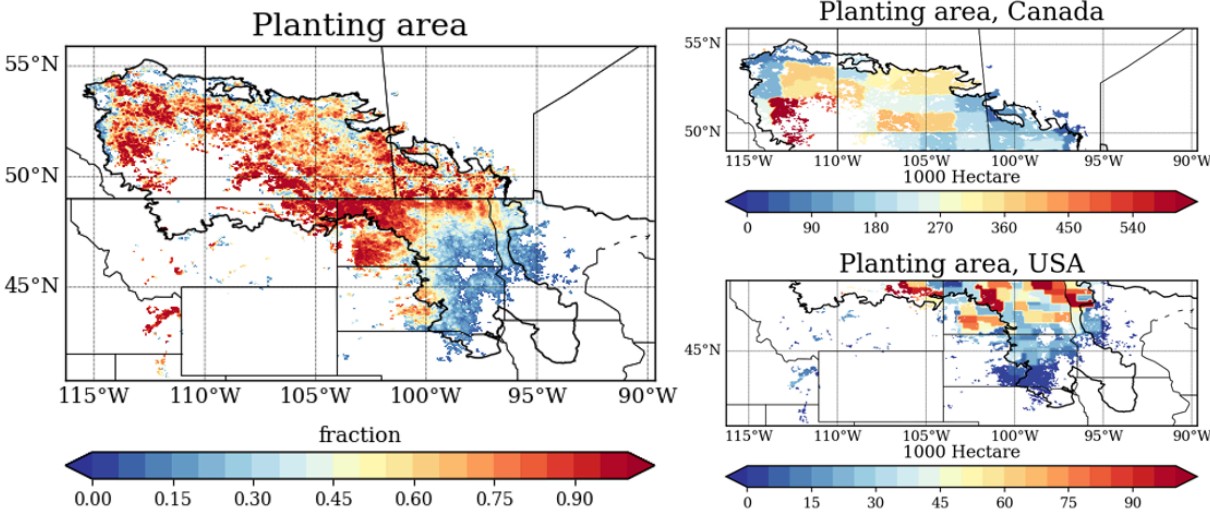

**Figure 3. Wheat planting fraction in the study area from the combined dataset from (a) GMU CropScape and AAFC ACI; (b)
Planting area in 1000 Hectares in the US North Great Plains and Canadian Prairies, two color scales are used to accommodate the
wide ranges of values from these two data sources.**


Yearly spring wheat yield data are collected from USDA-NASS and Statistics Canada, which are used for evaluation of the
modeled grain biomass. A unit conversion is necessary, considering the standard 15% of moisture content, from census yield
data (bu/ac in the US and kg/ha in Canada) to dry grain biomass ($g/m^2$) in model output, according to the test weight
conversion charts for Canadian grains (Canada Grain Commission).
To evaluate the wheat leaf phenology, the MOderate Resolution Imagine Spectroradiometer LAI product (MOD15A, Myneni
et al., 2015) is used, which provides regional LAI detection at 8-day temporal intervals and 1-km spatial resolution, starting
from 2001. This product provides the peak timing and spatial coverage of leaf dynamics within growing seasons, that are
useful for evaluating model crop phenology. The 8-day time slices from May 1 to Aug 29 are selected (16 time slices) to reveal
the wheat growing season in this study.



### 2.6 Experiments Design

In this study, three individual and progressing experiments are designed. This section provides a brief description of these experiments, and a summary table is presented (Table 1). Rainfed wheat is grown for most of the study region, except for isolated irrigated areas in southern Alberta and central Saskatchewan. Irrigation impacts on crop dynamics and yield will be the focus of future studies.

**Point-scale simulation:** The model is driven by the meteorological forcing collected at the Kenaston site, with prescribed planting & harvest dates. The purpose of this experiment is to obtain a set of wheat-specific parameters against the Kenaston site for three available site-years of observations. Site observations, including turbulent fluxes (sensible heat (SH) and latent heat (LH)), soil moisture, LAI, and aboveground biomass, are used to evaluate model performance. (See Table A1 in Appendix A for the full details of spring wheat growth parameters used in this study).

**Regional simulations:** To facilitate regional simulations, the meteorological forcing data were from the convection-permitting downscale WRF model simulation in the CONtiguous US (CONUS) and Southern Canada (Liu et al. 2016), spanning from 2000 to 2013. The advantages of this dataset are that the high-resolution grid spacing (4-km) provides detailed representation of the heterogeneous surface properties and it allows direction simulation of convective precipitation without using parameterization schemes, both of which are important to agricultural study (Prein et al., 2015; Li et al., 2019). The CONUS dataset has been widely used to study regional climate (Prein et al., 2016; Zhang et al., 2018) and hydrology (Zhang et al., 2020) in North America.

Two regional planting & harvest simulations are conducted using: (1) the province/state-level map for the most typical planting/harvest dates in Canada and U.S. (USDA NASS, 2010); or (2) dynamic dates for planting & harvest based on temperature in the growing season. These two simulations are compared to reflect dynamic agricultural management and interannual weather variability.

**Temperature stress function simulation:** The purpose of the third simulation is to test the wheat response to high temperature, and to demonstrate its variation among regional and interannual scales. This is done by replacing the default function from Eq. 6, Collatz et al. (1991), with a more realistic temperature stress function (Eq. 7) from Wang et al. (2017) (Figure 2b).

**Table 1. Experimental design for three sets of crop model simulations.**

| Experiment design | Location | Period | Purpose | Note |
|---|---|---|---|---|
| 1. Single-point simulation | Kenaston, SK | 2016, 2019 | Establish the single-point model and calibrate parameters | Model calibration and results evaluation |
| 2.Regional simulation | Northern Great Plains | 2001-2013 | 1.province/state-level planting/harvest 2. Dynamic planting/harvest | TAVE>10 for planting. and ACGDD>1500 for harvest. |
| 3. Temperature stress function | Northern Great Plains | 2001-2013 | Apply the temperature stress function for spring wheat | |




## 3 Results

### 3.1 Point-scale simulation

Figure 4 shows the growing season GDD, LAI, and aboveground biomass from the field observations and model results as

compared to the default (MODIS monthly climatology LAI) results. It is obvious that the new wheat model has better LAI dynamics compared to the default MODIS monthly LAI, especially for the timing and peak value of LAI.

The wheat growing season can be roughly divided into two stages, the vegetative stage and the reproductive stage (grain-filling). These two stages are separated by the time of peak LAI within the growing season, before which most of the assimilated carbon is allocated to leaf mass and after which to grain. Therefore, the grain biomass starts to accumulate after

the time of peak LAI, shown as the orange lines.

The Kenaston site only recorded the total aboveground biomass in 2019, without differentiation of biomass into individual plant organs. In Noah-MP crop, total aboveground biomass includes leaf, stem and grain and is simulated reasonably well for the two 2019 sites (data for 2016 were missing). The yields for 2019 were 2908 kg/ha and 1635 kg/ha for SW30 and SE13, where the model predicted 405 g/m$^2$ and 247 g/m$^2$, respectively, aligning with the higher end to the reported yields. These

higher yield estimates are discussed in section 3.3.

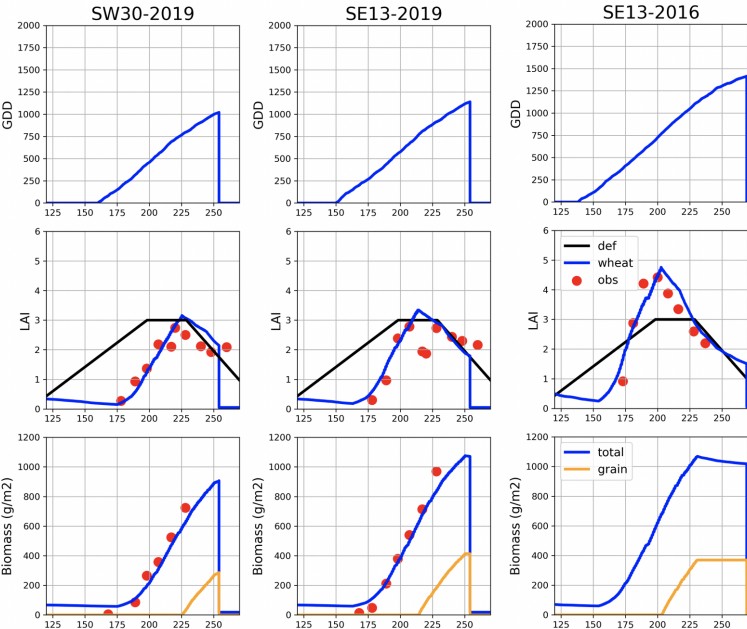

**Figure 4. Three site-year growing season GDD, LAI, and aboveground biomass for spring wheat from the field observation and the Noah-MP crop model.**


The turbulent fluxes (latent and sensible heat) and soil moisture, as daily time series, for the three growing seasons are provided in Figure 5. The simulated dynamic LAI from the wheat model presented higher latent heat fluxes compared to default LAI,





especially in three summer months (JJA). The soil moisture times series show that NoahMP crop consumes more soil moisture, especially from the second layer in late growing season since August, as it produces higher latent heat fluxes, suggesting more efficient turbulent exchange of water fluxes between soil and the atmosphere. This has great implications for the vast agricultural regions in the North America Great Plains, as the previous default model may have profoundly underestimated agricultural control of water feedback to regional weather and climate.

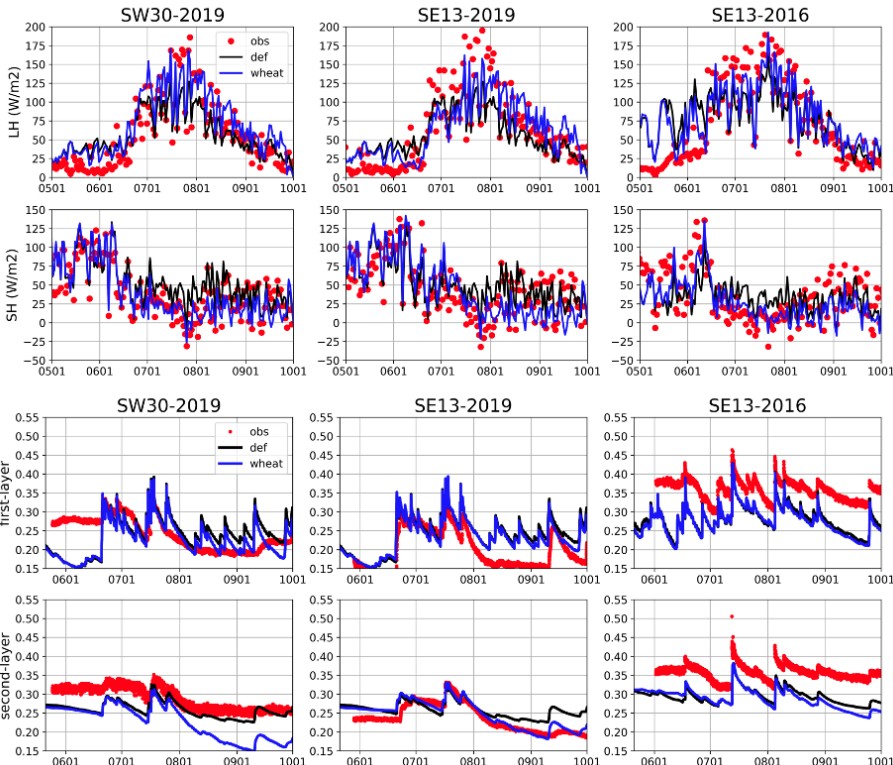

**Figure 5. Turbulent fluxes (sensible and latent heat, SH & LH) and soil moisture (two layers from 0~10cm and 10~40 cm) results from three site-year simulation.**

The Noah-MP crop model demonstrated reasonable simulations of soil moisture, crop growth, and turbulent fluxes transported to the atmosphere, facilitating the land-atmosphere coupling in croplands on the land surface branch. It is potential to further extending the coupling with WRF, such that could assess the crop growth feedback to regional climate.



## 3.2 Regional dynamics of LAI and yield

Figure 6 shows the spatial distribution of 8-day LAI time slices for the 2007 growing season, May 1st to Aug 29th (16 slices for 120 days, roughly 4 months). The time slices show that the regional LAI starts to green-up in late May, due to emergence, and reaches its peak around late July, entering the grain-filling stage. After that the LAI declines until crops mature. Finally,
the crops senesce prior to harvest at which time LAI drops to 0. Spatially, the U.S. portion of the growing region starts emergence and harvest generally earlier than the northwest portion in Canada, and its growing season is shorter - due to warmer average temperatures and faster heat accumulation.

As in the model, the province/state-level planting/harvest uses an arbitrary planting/harvest date for the most usual time windows, regardless of interannual weather variability and spatial heterogeneity within each state. Another obvious deficiency
of this approach is the obvious state/province boundaries in LAI values as shown in the figure. Even though two fields might be very closely located geographically, but are in different states/provinces, they show very different phenology as controlled by the state/province-level planting/harvest season. Even more so for the crop LAI across US and Canada border. These lead to the simulated spatial LAI homogenous within, but substantially discontinued across, a state or province.

The dynamic planting/harvest simulation substantially improves the regional LAI time slices within the growing season. The
TAVE threshold triggers planting earlier in the south, as temperature warms up earlier, and the GDD accumulation triggers earlier harvest as well, accounting for the higher/faster heat accumulation. The growing season in the northern region in Canada starts later and also extends longer. Considering these two dynamic thresholds accounts for the south-to-north transition in crop phenology.

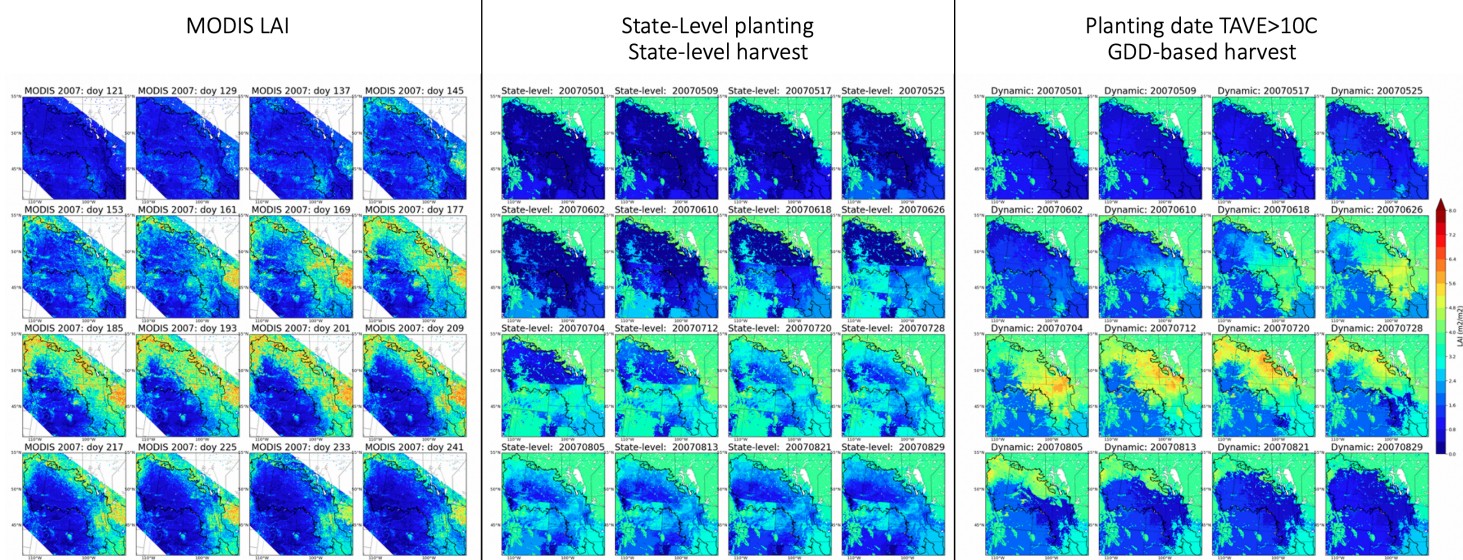

**Figure 6. 8-day LAI time slices from (a) MODIS product (MOD15A), (b) province/state-level planting/harvest and (c) the dynamic planting/harvest simulation.**





The large discrepancies between static and dynamic planting/harvest date simulation demonstrate its essential control of growing seasons on accumulated biomass. Figure 7 shows the spatial distribution of yields, planting and harvest date, and the

scatter plot between the grain-filling stage duration (days) and the final yield. The province/state-level grain-filling stage durations are much longer than in the dynamic simulation, thus leading to substantially overestimated yields compared to the dynamic planting/harvest simulation and census data. The color scale in the scatter plot indicates the planting date, which also suggests the earlier the planting, the longer the grain-filling stage, and hence, higher yield.

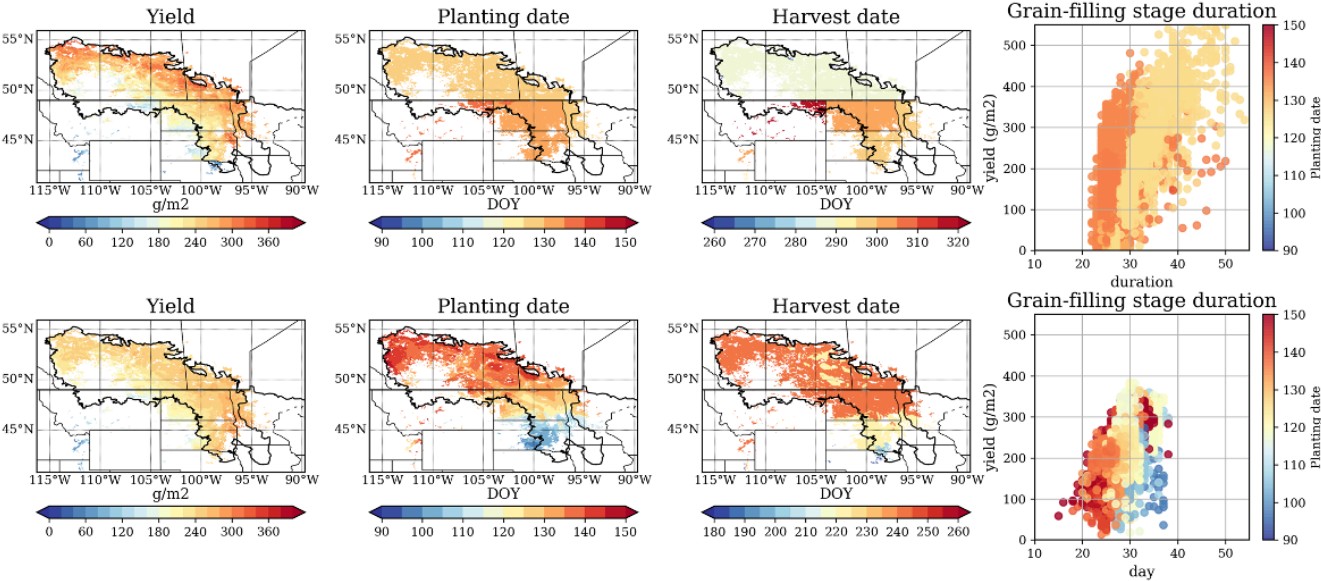

**Figure 7. Model results of (a) yield, (b) planting date, (c) harvest date and (d) scatter plot between yield (g/m2, y-axis) and duration of grain-filling stage (day, x-axis) from default province/state-level (top) and dynamic (bottom) planting/harvest and dynamic simulation.**





### 3.3 Temperature stress function

Figure 8 presents the spatial difference pattern between two simulations and the census data. Applying the temperature stress function in Wang et al. (2017) shows an obvious yield reduction around 10% from default simulations. The spatial distribution of this reduction is more evident in the southeast domain in North Dakota, Minnesota, and Manitoba, where the average temperature is higher and heat stresses are more likely. The yields in cooler provinces, Alberta and Saskatchewan, are less affected.

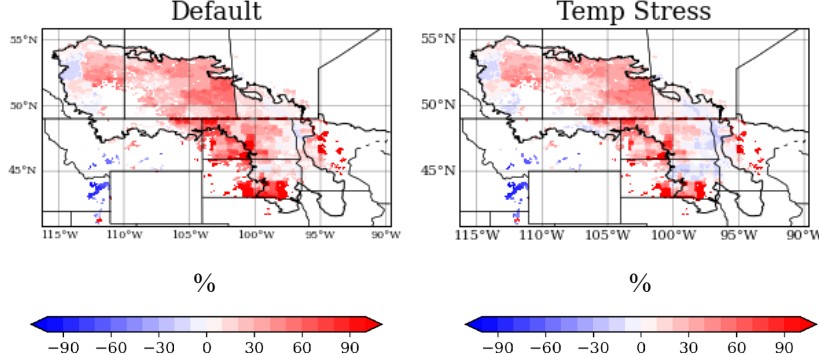


**Figure 8. Simulated grain yield compared to the census data from Statistics Canada and USDA-NASS from Default and Temp Stress simulation.**


The number of hot days (NHD) is a good indicator for the frequency of hot days (Zhang et al. 2011; Zhang et al., 2018). This index counts the frequency of days' maximum temperature exceeding a statistically defined P90 threshold for over 30 years. Figure 9 below shows the scatter plot relationship between the NHD within the growing season (May to Sep) and final yield in three Canadian Prairie Provinces.

Although presumably negative impacts of high temperature stress on wheat yield is expected, discrepancies among these three provinces are obvious in both census data and model results. For Alberta (AB) and Saskatchewan (SK), the negative impacts of extreme hot days on grain biomass are most evident and the model has captured this relationship reasonably well, suggesting that the temperature stress play a profound role in final crop yield. The temp stress simulation shows a further reduction on the final yield, especially with higher NHD, which contributes to less overestimated biases (about 10% reduction) compared to

the default configuration.

For Manitoba (MB), the higher quantile yield decreases with NHD while the lower quantile yields actually increase amid higher NHD. These two distinct features suggest that temperature stress is not the only factor limiting the final yield in this province. In some part of MB summertime precipitation is significantly higher than that in AB and SK, so that low NHD may suggest more precipitation and low exposure to sunlight, which in turns limits the photosynthesis and biomass accumulation.

In other words, the wheat productivity in MB is less water limited as those in AB and SK. These two distinct behaviors to NHD are also reasonably captured by the Noah-MP crop model and the temp stress modification restrain 10% final yield as well.



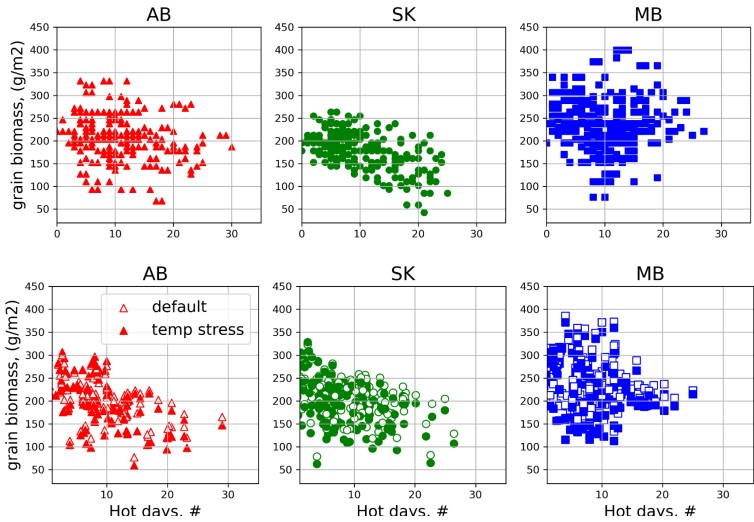

**Figure 9. Scatter plot of NHD (x-axis) vs yield (y-axis) at three Canadian Provinces, from Statistics Canada (top) and model results (bottom).**

From 2001 to 2013, an increasing trend of yield is apparent in each province, from both census data and model simulations (Figure 10). In addition, strong interannual variability exists in these trends, and the model's performance varies in each province. For MB, the model produces the best results in terms of trend and interannual variability. The temperature stress simulation further produces stressed yield results corresponding to heatwave events in 2002, 2006 and 2012, which largely agrees with the observations.

However, as for SK and AB, to capture the interannual variability is challenging, especially in AB, where the model shows a high yield peak while the census data shows a lower value. A possible speculation for this discrepancy would be that there are larger extends of irrigated croplands in AB than in other provinces, thus has less water limitation in those warmer years. Presumably, these irrigated high yields could bias the observed StatsCan yield, i.e., yield not connected to environmental conditions that the model is limited to. This suggests that although the model presents reasonable estimates of mean yield results, that it is very difficult to capture interannual variability, due to the weather variations within each growing season. Applying irrigation in NoahMP crop could potentially improve the crop yield simulated in AB, yet we currently lack the spatial distribution and irrigation amount data. These data are essential to improve our understanding of crop yield responses to heat and water stress and the model performance to reasonably capture such responses.



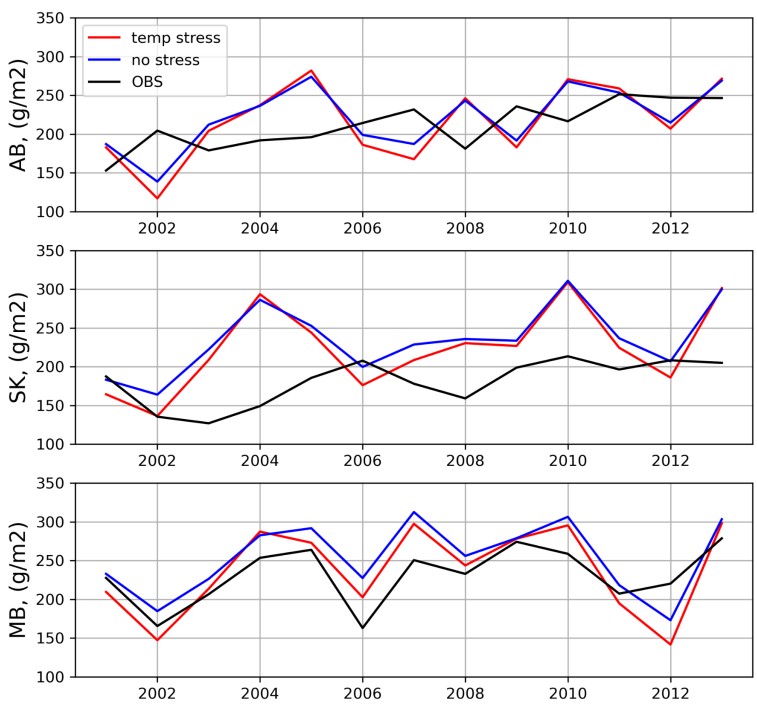

**Figure 10. Timeseries plot for these three provinces from 2001 to 2013.**

340

The statistics of yield from the census data and various simulations are presented in Table 2. As compared to the census data, the province/state-level simulation shows the highest yield among all simulations due to unadjusted growing season length and higher optimal temperature. The dynamic planting/harvest substantially restricts the yield in the northern part of the domain, as the province/state-level planting/harvest fails to represent the growing season dynamics in this region. Finally, the

345 temperature stress function cuts about 10% of the yield in Manitoba and North Dakota, while it has little effect in AB, where temperature stress is not as strong as other provinces.

**Table 2. Summary of the yield for different simulations over three regions.**

| Simulations (g/m$^2$) | AB | SK | MB&ND |
|---|---|---|---|
| Census data | 246.89 | 172.19 | 240.33 |
| State-level planting/harvest | 298.38 | 209.81 | 262.69 |
| Dynamic planting/harvest | 243.45 | 195.90 | 253.54 |
| Temp Stress | 243.89 | 192.46 | 231.18 |





## 4. Discussion

### 4.1 Discrepancies in using MODIS to evaluate growing season phenology

The MODIS 8-day LAI product provides the first remote sensing comparison for regional-scale modeled LAI for spring wheat in the Northern Great Plains. However, large uncertainties remain over its inadequacy in detecting different crop species and spatial resolution in representing the heterogeneity at field-scales. For example, there are three major crop types in Saskatchewan, cereals (mainly wheat and some barley), pulses (chickpea, lentils) and canola. However, with crop rotation between these types at scales within the MODIS LAI product, we cannot differentiate the LAI for specific crop types. For example, with a point-to-pixel comparison of the LAI time series from the Kenaston site with the MODIS LAI, higher values are shown from MODIS data, which could be related to canola in surrounding fields.

On the other hand, the original annual crop inventory data from AAFC and CropScape (at 30-m) only represents the temporal frequency of these crops - however it is often used as spatial fraction. It may be the case that for some places, wheat may be frequently planted, but not at significant extent compared to other crops types in the region. For the U.S. portion, corn and soybeans are also popularly planted in the Red river valley between Minnesota and North Dakota.

Due to the reasons above, previous regional crop model simulations reliant on large scale LAI remote sensing products in North America have challenges in modelling crop specific phenology and LAI dynamics. Thus, one should consider products like the MODIS LAI valuable to assess the qualitative evolution of LAI in growing seasons, while quantitative comparisons should be treated with caution and with an expectation of bias that relates to relative extent of the crop type of interest in the local crop rotation.

### 4.2 Uncertainty in Planting and Harvest Date

This study attempts to adopt the dynamic planting/harvest date as a more advanced approach to the province/state-level date. Yet there are much more complex dimensions involved in practice in reality. An aspect not captured by temperature/GDD thresholding is that there are differences in day length (and therefore incident shortwave radiation that drives photosynthesis/photoperiods) across this range in latitude that does moderate some of the GDD differences. Some crop models have already considered characterizing the day length (Setiyono et al., 2007) and can become common practice in also integrated ESM-crop model approach.

In the Canadian Prairies, the major restriction for planting and harvest is temperature, usually planting from late April to early May after sufficient time since snowfall and melting, while harvesting from late August to early September to avoid the first frost in fall.

Agronomic and logistical considerations determine the actual planting and harvesting dates to accommodate reality and non-optimal conditions. In cool/wet springs planting operations will be delayed until fields are dry enough to allow machine access, and seed placement will be shallow to take advantage of near surface soil moisture and facilitate faster emergence. In contrast, in warm dry conditions, planting will occur earlier with deeper seed placement to increase germination potential and to limit the period of soil evaporation losses and maximize crop water use efficiency. Harvest operations can also be disconnected





from physiological maturity as storage implications of grain moisture are critical to avoid post-harvest losses. Except for early
385     frost events, prior to maturity, quality losses after senescence are limited and so timing of harvest operations is determined by
the interactions between grain moisture and precipitation and humidity conditions.Then comes a critical question: given that
farmers keep planting/harvest in a quite stable time period while the year-to-year weather is fluctuating, whether or not to
incorporate these "in-depth" wisdoms into the modeling process? This question is still remained uncertain and unanswered in
current stage of crop model research.

390

**4.3 Temperature Stress Function**

In current crop models, heat stress impacts on crop growth and final yield in two ways: (1) is the temperature stress function
applied in the photosynthesis-stomata subroutine, with high temperature regulating/limiting the plant physiological function;
(2) is the required GDD threshold to progress into different growing stages, as high temperature induces faster heat
395     accumulation and shorter phenological developments, leading to yield loss. For the first way, multiple studies have dedicated
to address the first way – heat stress impacts on crop photosynthesis behaviors (Bernacchi et al., 2013; Siebert et al., 2014;
Levis 2014). For example, Siebert et al. (2014) claimed that the differences between applying canopy temperature and air
temperature in crop models under heat stress simulations. This highlights one of the advantages of integrated ESM- or LSM-
based crop models, such as NoahMP-crop, for using canopy temperature, calculated from energy balance, rather than air
400     temperature for photosynthesis process.

Moreover, high temperature stress has co-occurred along with water stress – rising temperature increases evaporation demand,
depleting soil moisture with dry conditions – lead to a compound heat-water stress (Lesk et al., 2022). As a result, previous
large-scale statistical studies (Lobell et al., 2011b) have revealed crop yield decline corresponding to temperature warmer than
30 ℃, which is 5~10 ℃ lower than plant-scale studies (Prasad et al., 2011). In our study, the new temperature stress function
adopted from Wang et al. (2017) has shown a lower temperature for optimal photosynthesis compared to the function
developed from Collatz et al. (1991) from lab measurements (35 ℃). Judging from the simulated results, the new temperature
stress function has better performance and less overestimation. To obtain a comprehensive understanding of temperature stress
on crop yield, we still call for potential applications of ESM-based crop models in the future, as temperature and moisture
processes are better integrated within model structures.

For the second way, there have been debates on the use of fixed GDD thresholds for regulating growing stages, especially
between crop model developers and genotype seed breeders. For example, crop modelers tend to use the GDD-based thresholds
to constrain and regulate crop growing seasons while these thresholds are artificial and empirical (based on the site where the
model is developed and is subject to parameter calibration). Recent advances through seed breeding and GMO's, have provided
new crop genotypes in every 2-3 years, which has been a fundamental reason for a continuous increasing trend in crop yield
in the last half century. For example, the recent effort in breeding for early-maturing and heat-tolerant wheat lines from South
Asia and adapt it in Mexico, by the International Maize and Wheat Improvement Center (CIMMYT) (Mondal et al., 2016).





This brings in a paradoxical situation that crop modelers, agroclimatic scientists, agronomists and farmers are relying on statics empirical GDD thresholds to estimate crop growth when these thresholds are in fact dynamic. So is there any value in using static GDD thresholds of current crops to predict future crop yield under climate change when forthcoming advances in

genotypes will almost surely break through such thresholds.

There is no simple yes or no answer to the question above. Nonetheless, such questions manifest themselves through the progress of model parameterization. Most crop modelling schemes that are capable of coupling crop-climate interactions such as NOAH-MP still require GDD-based approaches to characterize crop growth. Higher order representations that consider gene effects on phenology directly exist, but their complexity makes simulating the entire genetic controls on specific traits

impossible (Wang et al., 2019). In this context, it is critical to understand the GGD threshold approach limitations and consider approaches to capture the evolving GDD threshold dynamics.





## 5. Conclusion

Spring wheat production in the Canadian Prairies and U.S. Northern Great Plains is a significant source of wheat for domestic food supply and international exports. This study establishes an example for the development of a new crop species within the

Noah-MP crop model framework and its application in large regional simulations. The study further investigates the crop model responses to heat stress within a 13-year study period. We found the following:

1.      The point-scale spring crop model successfully captures spring wheat LAI dynamics for three site-years in Kenaston, SK, Canada, compared with default monthly climatology LAI. The simulated higher LAI results in more efficient water movement from the soil to the atmosphere, mediated by plants' stomata. Therefore, this single-point model demonstrates the

ability to quantify the vertical continuum of the complex energy-water-carbon exchange within the atmosphere-crop-soil system.

2.      To propagate the point-scale crop model to the regional-scale, dynamic planting/harvest triggers are applied to better depict the heterogenous farming practices than a province/state-level map used in a previous study. This approach not only improves the growing season LAI, as evaluated by MODIS 8-day LAI time slices, but also the final yield, compared to

agricultural census data. It is also shown that the modeled yield is closely related to the duration of crops' grain-filling stage, highlighting the importance of reasonably capturing the spring wheat phenology.

3.      An updated temperature response function is implemented within the photosynthesis-stomata process in the crop model to better represent the optimal temperature range for wheat growing conditions. This simulation shows reduced final yield by about 10% in southern states in the U.S. and Manitoba in Canada, while it does not have a significant effect in Alberta.

The interannual variability of crop yields is well captured for Manitoba, especially for the yield damages due to heatwaves in the recent decade.

Finally, the model results are discussed in various aspects, including limitations in uncertain planting/harvest dates and the heat stress function, as well as potential future development. The model's capability to reasonably simulate interannual

variability and the large spatial distribution of growing season LAI and final yields was demonstrated.

This work has great implications for developing the methods to address how crop production will be affected by future climate change, warmer temperatures, and uncertain precipitation patterns, which are critical to future research. be the next step in our research.



**Code Availability**

The Noah-MP model is driven by the NCAR high-resolution land data assimilation system (Chen et al., 2007) and can be downloaded from https://github.com/CharlesZheZhang/hrldas/tree/wheat (access through DOI: 10.5281/zenodo.7556048). The Noah-MP LSM can be accessed from https://github.com/CharlesZheZhang/noahmp/tree/wheat with the release of the code (v4.4 with spring wheat dynamics with DOI: 10.5281/zenodo.7556046)

**Data Availability**

The modeling results and analysis data used in this study is uploaded to and can be accessed by zenodo repository: https://doi.org/10.5281/zenodo.7023831.

The regional scale crop planting area data are available from the USDA/NASS website (https://nassgeodata.gmu.edu/CropScape/). The county-level crop yield data are available from USDA/NASS website (https://quickstats.nass.usda.gov/, last access: 2022 Oct).

The Census Agricultural Region data can be downloaded from Statistics Canada website (https://www150.statcan.gc.ca/t1/tbl1/en/tv.action?pid=3210000201, last access: 2022 Oct).

The 13-year forcing data for regional simulations are from the CONUS WRF simulation, can be accessed at https://rda.ucar.edu/datasets/ds612.0/TS1 (DOI: 10.5065/D6V40SXP).

**Author Contribution**

Zhe Zhang, Yanping Li, and Fei Chen conceptualized the study and Zhe Zhang performed the simulations. Zhe Zhang and Fei Chen designed the methodology, developed the model code and updated the model parameters. Phillip Harder and Warren Helgason provided the observational data resources for model validation. Zhe Zhang prepared the original draft of the manuscript with reviewing and editing from all co-authors. Yanping Li and Fei Chen provided valuable advice and supervision throughout the whole process.

**Acknowledgement**

Z. Zhang, Y. Li, P. Harder, W. Helgason, J. Famiglietti, and Z. Li thank the support from the Global Institute for Water Security and Global Water Futures Program by the Canada First Research Excellence Fund. F. Chen, P. Valayamkunnath, and C. He thank the support of the National Center for Atmospheric Research, Water System, USDA NIFA Grants 2015‐67003‐23460, NSF INFEWS Grant #1739705, and NOAA OAR Grant NA18OAR4590381. NCAR is sponsored by the National Science Foundation. Any opinions, findings, conclusions or recommendations expressed in this publication are those of the authors and do not necessarily reflect the views of the National Science Foundation.



**Appendix A**

Table A1. Spring wheat parameter table in the Noah-MP crop model in this study

| Parameter Name | Value | Unit | Physical meaning |
|---|---|---|---|
| PLTDAY | 145 | DOY | Planting date |
| HSDAY | 273 | DOY | Harvest date |
| GDDTBASE | 5 | ℃ | Base temperature for GDD accumulation |
| GDDTCUT | 30 | ℃ | Upper temperature for GDD accumulation |
| GDDS1 | 150 | Accumulated ℃ | GDD from seeding to emergence |
| GDDS2 | 450 | Accumulated ℃ | GDD from seeding to initial vegetative |
| GDDS3 | 770 | Accumulated ℃ | GDD from seeding to post vegetative |
| GDDS4 | 950 | Accumulated ℃ | GDD from seeding to initial reproductive |
| GDDS5 | 1120 | Accumulated ℃ | GDD from seeding to physical maturity |
| C3PSN | 1 | - | Indicator for C3 plant (1) or C4 plant (0) |
| KC25 | 30 | Pa | $CO_2$ Michaelis-Menten constant at 25 ℃ |
| AKC | 2.1 | - | Q10 base for KC25 |
| KO25 | 3E4 | Pa | $O_2$ Michaelis-Menten constant at 25 ℃ |
| AKO | 1.2 | - | Q10 base for KO25 |
| VCMX25 | 60 | umol $CO_2/m^2/s$ | Maximum rate of carboxylation at 25 ℃ |
| AVCMX | 1.5 | - | Q10 base for VCMX25 |
| BP | 1E4 | umol $/m^2/s$ | Minimum leaf conductance |
| MP | 9 | - | Slope of conductance-to-photosynthesis |
| QE25 | 0.06 | umol $CO_2$/umol photon | Quantum efficiency at 25 ℃ |
| Q10MR | 2.0 | - | Q10 base for maintenance respiration |
| DILE_FC_S5 | 0.5 | - | Coefficient for temperature leaf stress death |
| DILE_FC_S6 | 0.5 | - | - |
| DILE_FW_S5 | 0.2 | - | Coefficient for water leaf stress death |
| DILE_FW_S6 | 0.2 | - | - |
| FRA_GR | 0.2 | - | Fraction of growth respiration |
| LF_OVRC_S5 | 0.05 | - | Fraction of leaf turnover |
| LF_OVRC_S6 | 0.05 | - | - |
| RT_OVRC_S5 | 0.12 | - | Fraction of root turnover |
| RT_OVRC_S6 | 0.06 | - | - |
| LFMR25 | 0.8 | umol $CO_2/m^2/s$ | Leaf maintenance respiration at 25 ℃ |
| LFPT_S3 | 0.4 | - | Fraction of carbohydrate flux to leaf |
| LFPT_S4 | 0.3 | - | - |
| LFPT_S5 | 0.1 | - | - |
| STPT_S3 | 0.5 | - | Fraction of carbohydrate flux to stem |
| STPT_S4 | 0.6 | - | - |
| STPT_S5 | 0.2 | - | - |
| STPT_S6 | 0.2 | - | - |
| RTPT_S3 | 0.1 | - | Fraction of carbohydrate flux to root |
| RTPT_S4 | 0.1 | - | - |
| GRAINPT_S5 | 0.7 | - | Fraction of carbohydrate flux to grain |
| GRAINPT_S6 | 0.8 | - | - |
| BIO2LAI | 0.025 | $m^2$/kg | Leaf area per living leaf biomass |



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
