# Peer review of "Developing Spring Wheat in the Noah-MP LSM (v4.4) for Growing Season Dynamics and Responses to Temperature Stress"

_Geoscientific Model Development, 2022_

## Author Comment (AC1)

**Response to Referee** (#1)

**Ms. Ref. No: GMD-2022-311**

Title: Developing Spring Wheat in the Noah-MP LSM (v4.4) for Growing Season Dynamics and Responses to Temperature Stress
https://doi.org/10.5194/gmd-2022-311

We really appreciate the comments and questions posted by two reviewers during the open discussion. These comments and questions are very constructive and positive, which substantially help improve the manuscript. Please see our responses to two reviewers in blue texts presented in this document.

1. In section 2.3, it is important to note that the Y-axis in figure 2b represents V(T) and not f(TV), and the equation for the blue line is Wang 2017, not Wang-Engel (1998). Additionally, the plots in figure 2 should be labeled as a and b, and the Y-axis labels should match the text. Furthermore, the manuscript's novelty is the new heat stress function, but it is not adequately explained. Therefore, it is essential to provide a detailed scientific explanation of the function to help readers better understand it.

   We thank the reviewer for the advice on Figure 2 and highlights on our manuscript. We modified the Figure 2 with label (a) and (b) with title noting they are different responses for temperature and Vcmax function.

[Figure]

Figure 2 modified with label (a) & (b), and subtitles highlighting the responses for the temperature function and Vcmax (rubisco capacity) (also changed in the manuscript).

2. In Figure 3, it is recommended to include subplot numbers (a, b, c) to avoid confusion.

Thanks for the recommendation, subplot labels are added to Figure 3 in the revised version.

[Figure]

Adding subplot (a,b,c) for figure 3.

3. In line 190, it should be noted that "O" should not be capitalized in the MODIS abbreviation.

Thanks for the suggestion, we corrected this in the manuscript.

4. In line 206, the CONUS abbreviation's full form is incorrect; it should be "Conterminous" instead of "CONtiguous."

Thank you for the suggestion. Actually, both "conterminous" and "contiguous" are used in previous studies. For example, in Liu et al., (2016) at Climate Dynamics, the dataset was named as "contiguous United States" (CONUS). That's why we keep this name in the manuscript.

5. In line 219, it is necessary to clarify that there is no equation 7 in the manuscript and refer the readers to the comments in section 2.3.

Thanks for the reminder of equation 7. Indeed, it is critical and should be added to the manuscript.

6. In line 230, the sentence is ambiguous, and it is unclear what the author means. According to the given numbers, the model seems to have overestimated the yields. Therefore, it is recommended to rephrase the sentence to make it clear that the model overestimated the yields.

Thanks for the suggestion. We will rephrase the sentence: The model provides higher biomass estimate compared to the site recorded yield in 2019 (model: 405 g/m2 and 247 g/m2; site: 2908 kg/ha = 290.8 g/m2, and 1635 kg/ha = 163.5 g/m2).

7.  In section 3.1, it is essential to specify whether any other parameter tweaking was done during the single-point calibration and validation, besides GDD and Tavg for planting.

    This is an important suggestion and we appreciate it. The growth stage GDD parameters were adopted from a reference paper by Saiyed et al. (2008) showing the reference GDD for five sites across the Canadian Prairies.

    The growth stage carbon allocation parameters were adopted from the WOFOST model, for spring wheat: (https://github.com/ajwdewit/WOFOST_crop_parameters/blob/master/wheat.yaml). We will add this information in the manuscript.

8.  In Figure 6, the plot is overcrowded, and the colorbar is too small. To improve visibility, it is suggested to stack the three separate plots vertically and include a bigger horizontal colorbar. Furthermore, the description of Figure 6 needs improvement.

    Thanks for the suggestion, we changed the colormap according to the MODIS LAI image from NASA (from brown to green) and stacked the figure vertically. This helps better arrange the page.

[Figure]

LAI (m2/m2)

9. In Figure 9, it would be clearer to separately plot the default and temperature stress results. The figure will now have three rows. Also, due to the marker type, it is challenging to see the improvement during the temperature stress function.

We appreciate this comment and suggestion. To better demonstrate the temperature stress treatment, we add linear regression slopes in figure 9 for better visualization, for both observation and model results. The modified figure 9 is provided below with subplot labels.

The model results show obvious negative impacts of hot days on crop yield, but not as much in the observations, except in Saskatchewan. This implies that other factors, in addition to temperature, are playing a role. For example, irrigation is contributing to additional water sources in Alberta and wet springs requiring tile drainage in Manitoba. Applying the new temperature stress function reduces crop yield under high hot day numbers. This effect is stronger in AB and SK than in MB.

[Figure]

Figure 9 modified, with subplot labels. Scatter plots of numbers of hot days (x-axis) and grain biomass (g/m2) from observations (a-c) and model results (d-f). Linear regressions are also shown to better visualize the impacts of heat stress across provinces and two model simulations.

● It is necessary to provide a good scientific explanation of the P90 threshold and the temperature stress function being added.

The P90 temperature threshold for extreme temperature days is reviewed in Zhang et al. (2011: https://wires.onlinelibrary.wiley.com/doi/abs/10.1002/wcc.147) and Perkins & Alexsander (2013: https://doi.org/10.1175/JCLI-D-12-00383.1). This threshold is calculated from the long-term daily maximum temperature with 90th percentile. The daily maximum temperature exceeds which will be counted as a "hot day". In this way, the total number of hot days can be obtained throughout a growing season.

● In line 326 with the temperature stress simulation, the sentence claims that the stressed yield results correspond to heatwave events in 2002, 2006, and 2012, which mostly agree with the observations. However, it is not evident in Figure 10 that the temperature-stressed simulated yield matches the observations by comparing the overall

trend. For instance, in 2012, the default yield is much closer to the observed yields. Therefore, it is recommended to provide more clarity and detail.

Thank you for this comment. This sentence was trying to highlight the model performance on interannual variability of wheat yields in three provinces, being mostly captured in MB, but not so much in AB or SK. The temperature stress function reduces crop yields compared to the default function, especially for heatwave years, such as 2002, 2006, 2008 and 2011. For 2012, both default and the temperature stress simulation underestimate the crop yields, which suggests there might be other factors missing in the simulation.

- In line 397, "For example, Siebert et al. (2014) claimed that the differences between applying canopy temperature and air temperature in crop models under heat stress simulations." It is unclear what is being said about the differences between applying canopy temperature and air temperature in crop models under heat stress simulations. Therefore, the sentence needs to be rephrased to provide more clarity.

Thank you for pointing this out. We will rephrase the sentence: For example, Siebert et al. (2014) claimed there is substantial difference between when applying air temperature and canopy temperature in crop model studies during heatwaves - the latter can be 7$^\circ$C warmer than the former, depending on soil moisture.

- In line 437, it is necessary to mention whether the conclusion is for the region being studied. If the answer is yes, it is recommended to include this information in the sentence.

Thanks for the reminder. Yes, as suggested, we will include that the results obtained from this study are for the Northern Great Plains and Canadian Prairies. In general, as an effort to capture the spatial heterogeneity for the planting dynamics, introducing dynamic planting/harvest in some way would be encouraged to a larger region or global model crop model applications. We will add these descriptions into the manuscript.

---

## Author Comment (AC3)

**Response to Referee** (#2)

**Ms. Ref. No: GMD-2022-311**

Title: Developing Spring Wheat in the Noah-MP LSM (v4.4) for Growing Season Dynamics and Responses to Temperature Stress
https://doi.org/10.5194/gmd-2022-311

Crop growth and yield simulations in land surface models are crucial for both crop yield projections and the land-atmosphere interactions. Noah-MP is the latest generation of Noah land model. The authors added spring wheat growth (including dynamic planting/harvest and the temperature stress) into Noah-MP which improved its capabilities in application over the area where the spring wheat is the major crop. The manuscript is well written and there are only minor comments before considering for publication.

We gratefully thank the reviewers for their comments and suggestions, which we believe have substantially improved the original manuscript. The original reviewers' comments are reproduced below in black text and the corresponding response is shown in blue text.

Minor Comments:

Even though the authors mentioned that the 10 oC is based on a global synthesis of planting and harvest dates, the thresholds in equation 3 and 4 are quite arbitrary. I suggest to do some sensitivity analysis by adjusting the thresholds and show the current thresholds are reasonable for the region.

This is a great suggestion and we really appreciate it. We conducted sensitivity analyses on these two thresholds, TAVE and GDD, for one year (2007). The model results are compared with the weekly crop progress report provided by USDA, for planting and harvest (see figure below).

[Figure]

Cumulative distribution function (CDF) of the planting and harvest date from five parameter sensitivity analyses (TAVE=8, 9, 10, 11, 12 $^{\circ}$C for planting and GDD=1300, 1400, 1500, 1600, 1700 for harvest). The weekly progress report from USDA is also presented, with gray shaded areas indicating spatial variation for three states (North Dakota, South Dakota, and Minnesota).

Both parameters demonstrate strong influence on planting and harvest date in the North Great Plains region and indicate that average temperature and cumulative heat unit (e.g. GDD) have strong control of the management process on a large regional scale. These two figures show that the two parameters used

in our study (TAVE=10 and GDD=1500), obtained from the global synthesis of Sacks (2010), are reasonable.

In addition, we are currently developing a more comprehensive dynamic planting/harvest model, aiming to address the connections between interannual variability of growing season climate and planting/harvest date and areas. This model will involve not only average temperature and GDD, but also consider water availability, such as soil moisture. We will add this to the discussion of the manuscript.

For the regional simulation, the authors used 4-km resolution atmosphere forcing data. Could you prove the 4-km resolution yield a better LAI and yield simulation than other coarse resolution forcing, such as CRUNCEP? I'm curious for the regional averaged crop LAI and yield, will the coarse forcing showed a similar result as the 4-km high resolution.

Thank you for this great question. As for the CRUNCEP data, the dataset is 0.5° x 0.5° spatial resolution and 3-hourly temporal interval. We didn't perform simulations for coarse resolution forcing but it is an interesting question to explore for those who are interested.

We used the 4-km resolution forcing for two reasons: (1) the high-resolution convection-permitting forcing provides more reasonable precipitation, without using the convection parameterization scheme; (2) the high-resolution model grids better resolve surface representation, such as topography, land use, etc. These two advantages enable the model evaluation with available dataset, such as MODIS LAI at 500-m and USDA NASS crop yield at county-level.

With limited data (for example crop yield) or regional average analysis, the fine-resolution forcing may lose a lot of spatial details and prohibit the evaluation of high-resolution LAI data, as we shown in Figure 8 with the dynamic planting scheme. Thus, the advantages of high-resolution forcing don't stand out.

It is also worth noting that high-resolution dataset allows landowners and agencies to make decisions and policies. It is at this level where high-resolution forcing may provide more detailed information that coarse resolution data cannot. That's why we used 4-km forcing in the beginning.

In figure 5, adding the spring wheat model improved LH, but showed poor soil moisture simulations than the default model. Please comment on how to fix this problem in Noah-MP.
Thank you for this question. Figure 5 shows the new wheat model presents increased LH and reduced SH, but reduced soil moisture compared to default and observation. This could be due to two reasons. The first reason is the initialization of model states, such as soil moisture, snow in spring, etc, i.e. the model is drier than the observation when it starts. However, as the station was established in 2016 and a short period of data was used to initialize the model states. Therefore, discrepancies of soil moisture were found at the beginning of the simulation, in 2016 and SW30-2019. However, these discrepancies vary site by site - in 2019, SE13 site presents better soil moisture results compared to SW30, especially the second layer depth.
The second reason may be due to shallow groundwater aquifers in the Canadian Prairies. There is shallow groundwater less than a few meters deep in the Canadian Prairies, supplying unsaturated soil moisture during the growing season when the ET demand is strong. This shallow groundwater contributes nontrivially to the total water budget in the Canadian Prairies, but is particularly hard to represent in a

single-point model. We had a paper discussing the groundwater contribution to the region's water budget under current and potential future climate (Zhang et al., 2020: https://doi.org/10.5194/hess-24-655-2020), and we will add this information in the discussion of this manuscript.

---

## Referee Report (RR1)

**Comments**

I have thoroughly reviewed the manuscript "Developing Spring Wheat in the Noah-MP LSM (v4.4) for Growing Season Dynamics and Responses to Temperature Stress." All the comments have been incorporated well and improved the quality of this manuscript. Overall, it is a very good piece of work. The authors have conducted a comprehensive study that significantly contributes to understanding spring wheat growth and its response to temperature stress in the Noah-MP LSM. With a minor revision this paper is good to publish.

**Previous comment:**
In section 2.3, it is important to note that the Y-axis in figure 2b represents V(T) and not f(TV), and the equation for the blue line is Wang 2017, not Wang-Engel (1998). Additionally, the plots in figure 2 should be labeled as a and b, and the Y-axis labels should match the text. Furthermore, the manuscript's novelty is the new heat stress function, but it is not adequately explained. Therefore, it is essential to provide a detailed scientific explanation of the function to help readers better understand it.

**Follow up on the comment:**

With the added equations making the comparison between temperature stress function f(TV) and Rubisco capacity ($Vcmax25$) parameter responses to temperature V(TV) in default and old function is possible. TV is defined as canopy temperature. Default and modified f(TV) are represented by eq. 5 and 7, respectively; similarly, for V(TV), it is eq. 6 and 9.

**However, the respective figure (Figure 2 ) has different X and Y axis acronyms defined nowhere. The X-axis (f(TV) and V(TV)) and Y-axis (TV )degree Celsius) labels in Figure 2 should be consistent with the corresponding equations for consistency and clarity.**

---

## Author Response (AR2)

**Response to reviewer#1:**

**Comments**

I have thoroughly reviewed the manuscript "Developing Spring Wheat in the Noah-MP LSM (v4.4) for Growing Season Dynamics and Responses to Temperature Stress." All the comments have been incorporated well and improved the quality of this manuscript. Overall, it is a very good piece of work. The authors have conducted a comprehensive study that significantly contributes to understanding spring wheat growth and its response to temperature stress in the Noah-MP LSM. With a minor revision this paper is good to publish.

**Previous comment:**

In section 2.3, it is important to note that the Y-axis in figure 2b represents V(T) and not f(TV), and the equation for the blue line is Wang 2017, not Wang-Engel (1998). Additionally, the plots in figure 2 should be labeled as a and b, and the Y-axis labels should match the text. Furthermore, the manuscript's novelty is the new heat stress function, but it is not adequately explained. Therefore, it is essential to provide a detailed scientific explanation of the function to help readers better understand it.

**Follow up on the comment:**

With the added equations making the comparison between temperature stress function f(TV) and Rubisco capacity ($Vcmax25$) parameter responses to temperature V(TV) in default and old function is possible. TV is defined as canopy temperature. Default and modified f(TV) are represented by eq. 5 and 7, respectively; similarly, for V(TV), it is eq. 6 and 9.
**However, the respective figure (Figure 2 ) has different X and Y axis acronyms defined nowhere. The X-axis (f(TV) and V(TV)) and Y-axis (TV )degree Celsius) labels in Figure 2 should be consistent with the corresponding equations for consistency and clarity.**

We want to sincerely thank the reviewer for the time and effort in reviewing this manuscript. Indeed, most of the comments are very constructive and helpful to improve the quality of this paper. Please see the attached Figure 2, modified according to the above comment.

[Figure]

Figure 2. (a) Temperature response of the default Noah-MP PSN-stomata scheme (Collatz et al., 1991) and a new temperature response function (Wang-Engel., 2017) revised equation; (b) Rubisco capacity (Vcmax25) parameter responses to vegetation canopy temperature (TV).